# Treatment failure to sodium stibogluconate in cutaneous leishmaniasis: A challenge to infection control and disease elimination

Hermali Silva[1], Achala Liyanage[2¤a], Theja Deerasinghe[3¤b], Vasana Chandrasekara[4], Kalaivani Chellappan[5], Nadira D. Karunaweera[1]*

1 Department of Parasitology, Faculty of Medicine, University of Colombo, Colombo, Sri Lanka, 2 Base Hospital Tangalle, Tangalle, Sri Lanka, 3 District General Hospital Hambantota, Hambantota, Sri Lanka, 4 Department of Statistics & Computer Science, Faculty of Science, University of Kelaniya, Colombo, Sri Lanka, 5 Department of Electrical, Electronic and System Engineering, Faculty of Engineering and Built Environment, Universiti Kebangsaan Malaysia, Bangi, Selangor, Malaysia

¤a Current address: Department of Community Medicine, Faculty of Medicine, University of Ruhuna, Galle, Sri Lanka
¤b Current address: District General Hospital Embilipitiya, Embilipitiya, Sri Lanka
* nadira@parasit.cmb.ac.lk

**Data Availability Statement:** All the relevant data are within the paper and its Supporting Information files.

## Abstract

The first-line treatment for *Leishmania donovani*-induced cutaneous leishmaniasis (CL) in Sri Lanka is intra-lesional sodium stibogluconate (IL-SSG). Antimony failures in leishmaniasis is a challenge both at regional and global level, threatening the ongoing disease control efforts. There is a dearth of information on treatment failures to routine therapy in Sri Lanka, which hinders policy changes in therapeutics. Laboratory-confirmed CL patients (n = 201) who attended the District General Hospital Hambantota and Base Hospital Tangalle in southern Sri Lanka between 2016 and 2018 were included in a descriptive cohort study and followed up for three months to assess the treatment response of their lesions to IL-SSG. Treatment failure (TF) of total study population was 75.1% and the majority of them were >20 years (127/151,84%). Highest TF was seen in lesions on the trunk (16/18, 89%) while those on head and neck showed the least (31/44, 70%). Nodules were least responsive to therapy (27/31, 87.1%) unlike papules (28/44, 63.6%). Susceptibility to antimony therapy seemed age-dependant with treatment failure associated with factors such as time elapsed since onset to seeking treatment, number and site of the lesions. This is the first detailed study on characteristics of CL treatment failures in Sri Lanka. The findings highlight the need for in depth investigations on pathogenesis of TF and importance of reviewing existing treatment protocols to introduce more effective strategies. Such interventions would enable containment of the rapid spread of *L.donovani* infections in Sri Lanka that threatens the ongoing regional elimination drive.

## Introduction

Leishmaniasis is a vector borne disease caused by *Leishmania* parasites and transmitted by the phlebotomine sand flies [1]. Leishmaniasis in Sri Lanka is caused by *Leishmania donovani*

**Funding:** This study was supported by the University of Colombo (https://cmb.ac.lk/) under Grant Number AP/3/2/2017/PG/31 to HS; the National Institute of Allergy and Infectious Diseases (NIAID) of the National Institutes of Health (NIH), USA (https://www.niaid.nih.gov/) under Award Number U01AI136033 to NK. The content is solely the responsibility of the authors and does not necessarily represent the official views of the NIAID, NIH or the University of Colombo. The funders had no role in study design, data collection and analysis, decision to publish, or preparation of the manuscript.

**Competing interests:** The authors have declared that no competing interests exist.

zymodeme MON-37, with the disease manifesting predominantly as cutaneous leishmaniasis (CL) [2, 3]. CL has different clinical sub types such as papules, nodules, plaques and ulcers. First indigenous case of CL in Sri Lanka was reported in 1992, with a significant number of cases reported during an outbreak in the northern part of the country from 2001 to 2003 [4, 5]. A subsequent rise was seen in reported cases with two hot spots in the northern and southern parts of the country. By 2018, eight out of 25 districts in Sri Lanka had a recorded leishmaniasis incidence rate of more than 10 cases per 100,000 population with an estimate of nearly one third of the Sri Lankan population living with the risk of acquiring leishmaniasis [6].

As per the standard guidelines used by local clinicians, CL in Sri Lanka is treated with local infiltration of sodium stibogluconate (SSG) or cryotherapy with application of liquid nitrogen [7]. Sri Lanka has used intra-lesional SSG (IL-SSG) as a first-line treatment for CL over the past 2 decades. The average number of IL-SSG injections required for healing is estimated as 10 injections given at weekly intervals [8, 9]. SSG is used as a mainstay of treatment for leishmaniasis in many parts of the world [10, 11]. Observed *in toto*, drug resistance is a major emerging problem world over in chemotherapy of leishmaniasis, including visceral leishmaniasis (VL) in the Indian subcontinent (with *L. donovani* as the causative agent) [12–15]. Similar observations have been made in Sri Lanka with increasing numbers of CL cases failing to respond to regular treatment with IL-SSG [16, 17].

Sri Lanka is confronted with an atypical, predominantly dermotropic variant of *L. donovani*, which has the potential to visceralise, if the parasites by-pass the local tissue immune responses [2, 3, 18]. The parasite is genetically and biochemically closely related to VL-causing parasite in India (*L. donovani* MON-2) [3, 18]. Sand flies that belong to the *P. argentipes* complex are the vectors of leishmaniasis in Sri Lanka, and similar to a large extent to those reported from India, Nepal and Bangladesh [19–21]. Presence of genetically-related *Leishmania* parasites, *Phlebotomine* vectors, close geographical and social ties between Sri Lanka and other South Asian countries (with frequent travelling in between) are important factors to consider when decisions are made on patient management, disease control and elimination of leishmaniasis [22]. Many attempts have been made to achieve sustained elimination of VL in the Indian subcontinent but with limited success [23]. In Sri Lanka, there is a steady expansion in spatial spread of leishmaniasis over the past 2 decades with hotspots in transmission [24]. Persistent CL lesions despite treatment, due to poor-response or treatment failure to IL-SSG might be a contributory factor for such disease spread. The situation remains a challenge for the national public health sector, which also may be a threat for the elimination program in the South Asian region [5, 6, 25]. Such threats may further escalate with the ongoing national plans to make Sri Lanka a hub of tourism in the post-COVID era (https://www.sltda.gov.lk/en).

Treatment failure is a complex phenomenon and it is not synonymous with parasite drug resistance [26]. Multiple factors such as parasite drug resistance, drug related factors, host factors (viz. co-morbidities, host immune response, pharmacokinetics, nutritional status) and environmental factors can result in treatment failure [26, 27]. Cases of poor response to antimony have been previously reported from disease hotspots both in the north and the south of the country, particularly from the Anuradhapura and Hambantota districts respectively [16, 17]. Even though clinico-epidemiological characteristics of general CL patient populations have been described in the past, treatment failure in CL patients have caught only little attention in Sri Lanka [8, 28, 29]. Therefore, this study aimed to fill that void.

The aims of this study were to identify the clinical response pattern of CL patients to regular IL-SSG therapy, to describe the characteristics of treatment failed patients, their lesions and to understand possible factors associated with treatment failure in a selected cohort of patients in southern coastal region of Sri Lanka. The findings of this study provided baseline information

on therapeutic response of *L. donovani* causing CL in southern Sri Lanka following treatment with IL-SSG and possible factors associated with treatment failure that are important in making policy decisions related to treatment modalities recommended for CL patients.

## Materials & methods

### Ethics statement

This study has been approved by the Ethics Review Committee, Faculty of Medicine, University of Colombo, Sri Lanka (EC-16-080). Written, informed consent was obtained during prospective data collection. Data were analysed anonymously.

### Study sites, design and participants

A descriptive cohort study was conducted in Hambantota District, in Southern Province, Sri Lanka, at the District General Hospital Hambantota (6.1268$^0$N, 81.126$^0$E) and Base Hospital Tangalle (6.02278$^0$N, 80.7975$^0$E), from April 2016 to October 2018. Data of CL patients who attended the dermatology clinic during the study period were obtained both retrospectively through their clinic records and prospectively through follow-ups. The southern region was selected as the study area as it forms the biggest hotspot of leishmaniasis transmission in the country [6]. The two selected hospitals are located in the area (Hambantota district) that accounts for the majority of CL patients reported from the southern region to the central database, maintained by the Ministry of Health (www.epid.gov.lk).

A total of 201 laboratory-confirmed CL patients were recruited to the study, adhering to the selection criteria: 1) patients who were treated with IL-SSG and were aged between 1 to 70 years, 2) with less than 5 lesions 3) longest diameter of the lesion less than 5 cm and 4) sought treatment within 1 year of lesion onset. Patients were treated by the clinicians of the study sites with IL-SSG [Stibovita$^{TM}$, Vital Healthcare Pvt. Ltd., India], each 1ml containing sodium stibogluconate BP 330mg (which is equivalent to 100mg of pentavalent Antimony)]. *Treatment schedule*: similar schedule was used on all patients irrespective of the age categories in accordance with the standard guidelines in Sri Lanka and carried out by the same clinicians in the two study sites. One to 3 ml of SSG (i.e., maximum dose of 300mg of pentavalent Antimony per week), as intra-lesional injections until the lesion was fully infiltrated, was given once a week. The lesions were assessed for clinical signs of healing as per the study definition, after 1 week of the last IL-SSG injection, weekly until the patient achieved complete healing or completed 10 IL-SSG injections, after which the patients were relieved from the study. Case definitions used in this study were: 1) '*Completely healed by IL-SSG*': A patient who has been treated only with IL-SSG weekly injections and has achieved complete cure as per established criteria, i.e.: complete flattening of papules, nodules or plaques and no open ulcer or induration or any sign of inflammation, with treatment of 10 or a lesser number of IL-SSG weekly injections, 2) '*Treatment failure*': A patient whose CL lesion was not completely healed following at least 10 injections of weekly IL-SSG or if the patient has had a relapse of the lesion after previous complete course of IL-SSG treatment (10 or more weekly injections) [17]. Treatment failures were managed at the clinic as decided by the treating clinicians.

The age, gender, number of lesions on a patient, site of lesion, type of lesion at the time of first visit to the dermatology clinic and the time duration since the onset of the lesion to starting treatment with IL-SSG were recorded. Tissue scrapings, tissue aspirates and/or tissue biopsies were obtained from active edges of skin lesions from new clinic attendees and used for laboratory confirmation of CL by microscopic examination of *Leishmania* amastigotes and/or *in-vitro* culturing of *Leishmania* promastigotes and/or by histopathology. Laboratory confirmation reports were traced for those who have been diagnosed previously.

## Statistical methods

Patients' data were expressed as 'mean ± standard deviation' or as a percentage with respect to the total number. Two association tests, Kruskal-Wallis test for scale variables and Chi-squared test for categorical variables, were carried out to check the association between the response variable 'Response to IL-SSG' and the predictor variables ($H_0$: There is an association between the two variables, $H_1$:There is no association between the two variables). Binary logistic regression analysis was carried out to find the factors associated with treatment failure (S1 Table). A 'Final regression model' was formulated as a binary logistic regression with backward Wald, using the variables: categorised *time since onset (TSO)*, categorised *age*, other variables in their natural form and taking treatment failure as the base category. The 'Final regression model' was fitted for each of the age categories 1–30 years (65 cases), 31–50 years (75 cases) and 51–70 years (61 cases), where the classes were approximately balanced.

## Results

Total study population was 201 patients and the selected variables viz. age, time since onset to seeking treatment and number of lesions on a single patient, were comparable between the 'completely healed' and 'treatment failure' groups (p-values > 0.05) (Table 1, S1 Datasets).

The total study population was predominantly males (63.7%), most lesions were on shoulder and upper limbs (43.3%) and most patients had sought treatment 2.1 to 4.0 months since

**Table 1. Characteristics of cutaneous leishmaniasis patients recruited for the study and comparison between the 'completely healed' and 'treatment failure' groups.**

| Factor | Total study population (n = 201) | Treatment outcome after 10 weekly IL-SSG injections | | Completely healed group versus Treatment failures |
| --- | --- | --- | --- | --- |
| | | Completely healed (n = 50) | Treatment failures (n = 151) | p-value[#] |
| Age (years)[*] | 38.1±17.5 | 35.6±19.2 | 39.0±16.9 | 0.240 |
| Age category (years)[*] | | | | |
| 1–10 | 6.5±3.2 | 6.2±4.4 | 6.6±2.2 | 0.806 |
| 11–20 | 14.9±2.8 | 14.8±3.0 | 14.9±2.9 | 0.937 |
| 21–30 | 27.4±2.4 | 27.6±2.5 | 27.3±2.5 | 0.812 |
| 31–40 | 35.9±3.2 | 34.7±3.4 | 36.2±3.2 | 0.208 |
| 41–50 | 45.0±3.1 | 44.3±2.3 | 45.2±3.4 | 0.399 |
| 51–60 | 55.6±3.0 | 56.8±3.4 | 55.3±2.9 | 0.227 |
| 61–70 | 65.4±3.0 | 66.2±3.5 | 65.2±3.0 | 0.555 |
| Time since onset (months)[*] | 4.1±2.8 | 4.3±3.2 | 4.0±2.6 | 0.612 |
| Time since onset categories (months)[*] | | | | |
| 0–2.0 | 1.5±0.5 | 1.7±0.5 | 1.5±0.5 | 0.633 |
| 2.1–4.0 | 3.3±0.5 | 3.4±0.6 | 3.2±0.5 | 0.331 |
| 4.1–6.0 | 5.5±0.5 | 5.6±0.5 | 5.5±0.5 | 0.716 |
| 6.1–8.0 | 7.5±0.6 | 7.0±0.0 | 7.6±.6 | 0.181 |
| 8.1–10.0 | 9.7±0.6 | 10±0.0 | 9.5±0.7 | 0.667 |
| 10.1–12.0 | 11.9±0.3 | 11.8±0.4 | 12.0±0.0 | 0.255 |
| Number of lesions on a single patient[*] | 1.2±0.6 | 1.4±0.7 | 1.2±0.5 | 0.052 |

Note

[*]Data expressed as 'mean ± standard deviation'.

[#] Independent-Samples t test was performed to calculate the p-values.

**Table 2. Characteristics and frequencies of laboratory-confirmed CL patients who were treated with IL-SSG weekly injections (n = 201).**

| Factor | Number (%) of study subjects (Total n = 201) | Number (%) study subjects in study groups | | Dummy variable names used for the regression analysis |
|---|---|---|---|---|
| | | Completely healed (Total n = 50) | Treatment failures (Total n = 151) | |
| **Age category (years)**[**] | | | | |
| 1–10 | 19 (9.5%) | 8 (16.0%) | 11 (7.3%) | Age 1 |
| 11–20 | 18 (9.0%) | 5 (10.0%) | 13 (8.6%) | Age 2 |
| 21–30 | 28 (13.9%) | 5 (10.0%) | 23 (15.2%) | Age 3 |
| 31–40 | 43 (21.4%) | 9 (18.0%) | 34 (22.5%) | Age 4 |
| 41–50 | 37 (18.4%) | 11 (22.0%) | 26 (17.2%) | Age 5 |
| 51–60 | 37 (18.4%) | 8 (16.0%) | 29 (19.2%) | Age 6 |
| 61–70 | 19 (9.5%) | 4 (8.0%) | 15 (9.9%) | |
| **Time since onset categories (months)**[**] | | | | |
| 0–2.0 | 58 (28.9%) | 16 (32.0%) | 42 (27.8%) | TSO 1 |
| 2.1–4.0 | 74 (36.8%) | 18 (36.0%) | 56 (37.1%) | TSO 2 |
| 4.1–6.0 | 42 (20.9%) | 8 (16.0%) | 34 (22.5%) | TSO 3 |
| 6.1–8.0 | 12 (6.0%) | 2 (4.0%) | 10 (6.6%) | TSO 4 |
| 8.1–10.0 | 3 (1.5%) | 1 (2.0%) | 2 (1.3%) | TSO 5 |
| 10.1–12.0 | 12 (6.0%) | 5 (10.0%) | 7 (4.6%) | |
| **Gender**[**] | | | | Gender |
| Male | 128 (63.7%) | 31 (62.0%) | 97 (64.2%) | |
| Female | 73 (36.3%) | 19 (38.0%) | 54 (35.8%) | |
| **Lesion site**[**] | | | | |
| Head or neck | 44 (21.9%) | 13 (26.0%) | 31 (20.5%) | Site 1 |
| Trunk | 18 (9.0) | 2 (4.0%) | 16 (10.6%) | Site 2 |
| Shoulder & upper limbs | 87 (43.3%) | 19 (38.0%) | 68 (45.0%) | Site 3 |
| Lower limbs | 40 (19.9%) | 10 (20.0%) | 30 (19.9%) | Site 4 |
| Multiple sites | 12 (6.0%) | 6 (12.0%) | 6 (4.0%) | |
| **Lesion type**[**] | | | | |
| Papule | 44 (21.9%) | 16 (32.0%) | 28 (18.5%) | Type 1 |
| Nodule | 31 (15.4%) | 4 (8.0%) | 27 (17.9%) | Type 2 |
| Plaque | 27 (13.4%) | 6 (12.0%) | 21 (13.9%) | Type 3 |
| Ulcer | 96 (47.8%) | 23 (46.0%) | 73 (48.3%) | Type 4 |
| Multiple types | 3 (1.5%) | 1 (2%) | 2 (1.3%) | |

Note: Data expressed as 'number of cases (percentage with respect to the total number n)'

the onset of the lesions with the majority (65.7%) obtaining treatment within the first 4 months since the onset (Table 2, S1 Datasets). Most study participants were adults viz. 31–40 years = 21.4%, 41-50years = 18.4%, 51-60years = 18.4% with a lesser proportion of young adults 21-30years (13.9%) in that category. Most common lesion type was ulcers (96/201, 47.8%).

In the total patient group, 151 patients (151/201, 75.1%) failed treatment with IL-SSG with persistent lesions, unlike the balance whose lesions healed completely (50/201, 24.9%). Treatment failure (TF) was more in the age group 21–70 years (127/164 = 77%) than in the 1–20 year olds (24/37 = 65%). Lesions on trunk showed highest TF (16/18, 89%) while those on head and neck showed the least (31/44, 70%). Of the four clinical types, nodules were least sensitive to treatment (TF: 27/31, 87.1%) while papules were the most sensitive (TF: 28/44,

**Table 3. Results of association tests for predictor factors and the treatment response to IL-SSG.**

|  | *Age | #Age category | *Time since onset | #Time since onset category | *No of lesions on a patient | #Gender | #Lesion site | #Lesion type |
|---|---|---|---|---|---|---|---|---|
| p-value | 0.361 | 0.541 | 0.968 | 0.644 | 0.044 | 0.775 | 0.146 | 0.219 |

Note

*Kruskal–Wallis test.

# Chi-squared test.

63.6%). The proportion of patients in the 61–70 age group of study population was comparatively low (9.5%) and the proportion of TF in this age group was higher (15/19, 9.9%) than those who have healed completely (4/19, 8.0%).

Application of association tests in statistics showed a statistically significant association only between the number of lesions and the treatment response (p-value = 0.044) and all other factors had p-values greater than 0.05 (Table 3).

Since more treatment failures were seen among adults (21–70 years) than in children and adolescents aged 1 to 20 years (77% versus 65%), the direction of the analysis was shifted towards age category-wise treatment response (Fig 1).

There were 3 trends noted in treatment failure viz. 1) increase in treatment failure from age 1–30 years, 2) decrease in treatment failure from age 31–50 years and 3) increase in treatment failure from 51–70 years (Fig 1, S1 Datasets).

When the 'Final regression model' was fitted for each of the above three age groups, the equation of the fitted final regression model for the age group 1–30 years was as follows:

$$\log\left(\frac{p}{1-p}\right) = -1.714 + 2.105(NoOfLesions) + 1.825(Site1) - 3.063(TSO1)$$
$$- 2.535(TSO2) - 3.551(TSO3)$$

The outcome of the modelling experiment indicated that the time since onset ≤ 6 months was significantly associated with increase in treatment failure while multiple lesions and

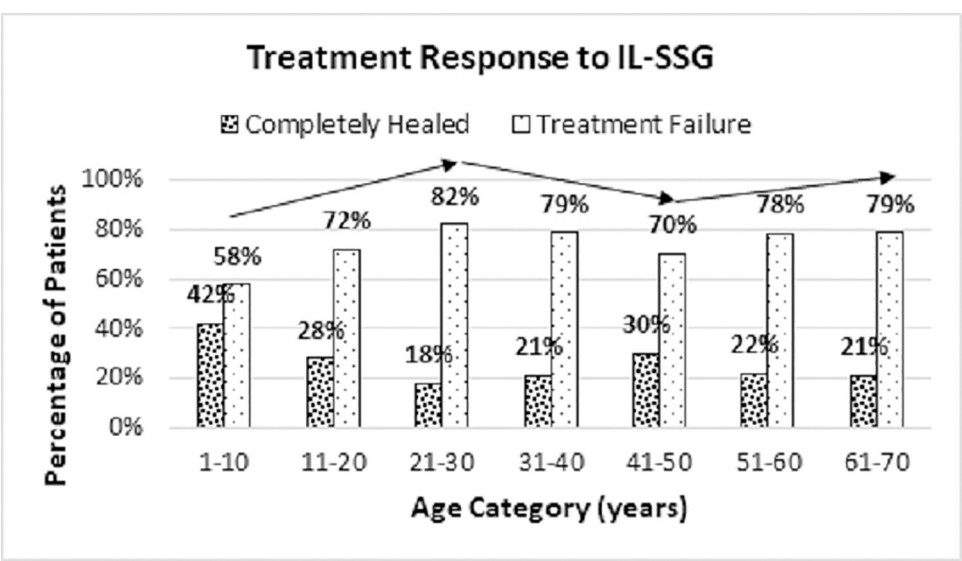

**Fig 1. Treatment response to IL-SSG with age.**

location of lesion on head or neck were significantly associated with decrease in treatment failure. For the other two age categories, none of the variables were preserved in the final model. Considering the order of variables removed from the model and their coefficients, the inference was that in the age group 31–50 years, nodules and location of lesions on sites other than the head and neck would create an impact on treatment response to increase treatment failure, while time since onset ≤ 2 months would have the opposite effect to decrease treatment failure. In the age group 51–70 years, only the time since onset ≤ 4 months seemed to reduce the likelihood of treatment failure (S1 Table).

## Discussion

Leishmaniases is a group of diseases which includes several clinical forms. To understand and counter the emerging drug resistance, it is essential to investigate into the different disease forms based on their clinical and serological features, to enable prompt diagnosis and effective management (Table 4) [30–42].

Antimony has been extensively used for the treatment of CL and VL for more than half a century. Emerging antimony resistance of CL (the most common clinical type) and VL (the most severe form) is a challenge against successful implementation of strategies for the control and elimination of leishmaniasis. Antimony failure has been observed more following its parenteral use in VL patients than when used intra-lesionally in CL patients or when combination of systemic and intralesional treatment is used (though the difference was not statistically significant) [43]. The extensive antimony resistance in VL patients in Bihar, India has even led to changes in treatment policy [43]. However, pharmacokinetics of antimony administered either IV or IM are similar [44]. Thus, the mechanisms of antimony resistance appear multifaceted, the knowledge of which remains important to combat treatment failure. Particularly important are the molecular mechanisms which are important in discovering targets for new drugs. Commonly discussed antimony resistant mechanisms are summarised in Table 5.

Participants of this study were from Hambantota district, in the southern leishmaniasis hotspot which has recorded the highest caseload from 2001 to 2018 [6]. A high proportion of the study population (75.1%) failed to achieve complete healing with standard treatment of IL-SSG. Proportion of treatment failures in the age group 21–70 years (127/164 = 77%) was higher than in the 1–20 year olds (24/37 = 65%). Diverse factors associated with or having an impact on treatment failure were observed among the age groups 1–30 years, 31–50 years and 51–70 years.

Treatment response is largely dependent on host's immune response [63] with the latter influenced by the age of the patient. Immune ageing, which occur with increasing age, would result in decreased activation of macrophages to kill intracellular parasites and reduce direct cytotoxic killing of parasitized cells by the NK cells [64, 65]. Additionally, the effective immune response that supplements the anti-leishmanial properties of SSG is likely to be reduced with age. These would contribute to the increase in treatment failure with advancing age as observed in this study. Furthermore, the increasing trend in treatment failure from children to adolescents and the reversed trend in adults from 31–50 years (Fig 1), could be due to the percentage decrease of CD4+ T cells from infancy until adolescence and rising trend observed thereafter [64]. CD4+ T cells are known to be important for Th1 proinflammatory response which promotes parasite clearance and healing.

Age-related thickening of dermis in children and adolescents, in contrast to its thinning in adults (with dermal atrophy occurring beyond 50 years of age) have been documented earlier [66]. Such age-related phenomena could reduce the establishment of parasites in dermal macrophages that may explain the low frequency of CL in older people. Similarly, reduce the

**Table 4. Characteristics of common clinical forms of Leishmaniases.**

| | Clinical form, the causative parasite/s and prevalence | Clinical profile and serological diagnosis | Ref |
|---|---|---|---|
| **(1)** | **Visceral leishmaniasis (also known as kala-azar)** <br> *Leishmania donovani* complex: <br> • Indian subcontinent and Africa—*L. donovani* <br> • Mediterranean basin, Central and South America—*L. infantum* (also known as *L. chagasi*) <br> *Prevalence* <br> • Mostly reported from Brazil, East Africa and India <br> • More than 90% of global VL cases in 2019 occurred in 10 countries: Brazil, India, Ethiopia, Eritrea, Iraq, Kenya, Nepal, Somalia, South Sudan, Sudan | Systemic and most severe form. Fatal if untreated. Irregular, long-term fever often associated with rigor and chills. Splenomegaly. Hepatomegaly. Lymphadenopathy. Pancytopenia. Anaemia. Weight loss. <br> *Serological diagnosis* <br> • Strong humoral response <br> • Antibody detection by—Direct Agglutination Test (DAT), Enzyme-linked Immunosorbent Assay (ELISA), Indirect Immunofluorescence Antibody Test (IFAT), rapid immunochromatographic rK39 dip-stick testing <br> • rK39 appears to be more sensitive in Asia than in Africa because the humoral response is stronger in most Asian countries resulting in higher antibody levels. <br> • Antibody testing cannot differentiate between active, past, sub-clinical, relapse and re-infections. <br> Antigen testing: Antigen-based latex agglutination test (KAtex) has shown good specificity in detecting antigen in urine with low to moderate sensitivity in Asia and Africa. | [30, 31, 35–38] |
| **(2)** | **Cutaneous leishmaniasis** <br> • Old World (the Eastern Hemisphere): *L. aethiopica, L. donovani, L. infantum, L. major,* and *L. tropica.* <br> • New World (the Western Hemisphere): Two major subgenera, *Leishmania Leishmania* and *Leishmania Viannia* <br> *Prevalence* <br> • CL is more widely distributed <br> • About 95% of cases in 2019 were from the Americas, Central Asia, Mediterranean basin and Middle East. | • Most common form. Cause skin lesions. <br> • Old World CL: Starts as a papule leading to an ulcer, usually 1–2 lesions but rarely multiple, heal spontaneously with disfiguring scars. <br> • New World CL: localized skin lesions <br> • A rare, chronic CL called lupoid/ recidivans/ relapsing CL is caused by *L. tropica* <br> *Serological diagnosis* <br> • Poor humoral response. <br> • rK39 rapid kit–negative <br> • Old world: serological testing not useful <br> • New world: IFA and ELISA with *L. amazonensis* antigen shown to be effective in diagnosing *L. braziliensis* and *L. guyanensis* infections <br> • Serology cannot differentiate past and active infections. | [30, 31] |
| (3) | **Mucocutaneous leishmaniasis** <br> Commonly caused by New World species <br> • New World: *L. braziliensis, L. panamensis,* and *L. guyanensis* <br> • Old World: *L. donovani, L. major, and L. infantum* <br> *Prevalence* <br> 90% of cases reported from Brazil, Peru, Bolivia and Ethiopia | Destructive lesions of nasopharyngeal mucosa <br> *Serological diagnosis* <br> • rK39 rapid kit: positive band <br> • Serological testing may aid the diagnosis in the presence of other indicators such as clinical and histopathologcal features <br> • A rising titre of antibodies indicate a relapse. | [30, 31, 39] |
| (4) | **Diffuse cutaneous leishmaniasis** <br> • *Old World*: *L. aethiopica,* to a lesser extent *L. major* <br> • New World: Species of subgenus *Leishmania*: *L. mexicana* and *L. amazonensis* <br> • No reports of *Viannia* causing diffuse-CL to date <br> *Prevalence* <br> • Old World: Uncommon <br> • New World: South and Central American countries | A rare syndrome. Chronic, multiple, non-ulcerative, lepromatous lesions spread over the whole body except on scalp, axillae, inguinal folds, palms and soles. Plaques are common but papules, nodules, macules and erythema may also be seen. No mucosal involvement. Do not heal spontaneously. Poorly respond to treatment and frequently relapse after treatments. <br> *Serological diagnosis* <br> • Strong humoral response <br> • rK39 rapid kit: strong positive band | [30] |
| (5) | Disseminated cutaneous leishmaniasis <br> New World: *L. brazilliensis, L. Mexicana* <br> *Prevalence* <br> Mainly found in the New World and rarely from Old World | Co-existence of different types of lesions such as papules, nodules & ulcers. Ulceration is common. Plaques are rare. May have mucosal involvement. Not chronic. Better response to treatment than diffuse-CL. <br> *Serological diagnosis* <br> • Strong humoral response <br> • rK39 rapid kit: strong positive band | [40, 41] |
| (6) | **Post-kala-azar dermal leishmaniasis (PKDL)** <br> Late complication of VL caused by *L. donovani.* Can be seen rarely in patients infected with *L. infantum* <br> *Prevalence* <br> Predominantly occur in South Asia and East Africa. | • Majority occur after recovering from VL but may occur without previous VL or simultaneously with VL. Interval between VL and PKDL in Asia is more (0 to 3 or more years) than in Africa (0 to 13 months). Clinical features differ in Asia and Africa. <br> • Asia: Macular rash is more common, and treatment is indicated. <br> • Africa: Papular rash is more common, spontaneously heal and treatment is indicated only for chronic or severe cases <br> *Serological diagnosis* <br> • rk39, DAT, ELISA are usually positive but is of limited value in diagnosis due to remaining serological response due to VL <br> • Serology is helpful in differentiating from other diseases such as leprosy | [30, 31, 40] |

(*Continued*)

**Table 4.** (Continued)

| | Clinical form, the causative parasite/s and prevalence | Clinical profile and serological diagnosis | Ref |
|---|---|---|---|
| (7) | *Leishmania*-HIV coinfection <br> Different causative species have been reported and some have shown atypical presentations such as dermotropic species causing VL in HIV-infected patients. <br> *Prevalence* <br> Highest prevalence had been reported in South-western Europe, but the numbers have been increasing in sub-Saharan Africa, South Asia and Brazil. | Leishmaniasis is an important opportunistic infection in HIV-infected patients. VL is commonly associated with HIV but association of other clinical forms such CL and MCL have also been reported. Clinical features could be similar to the classical clinical form, more severe or atypical. HIV and *Leishmania* infection reinforce each other. <br> *Serological diagnosis* <br> • Antibody based tests are less sensitive and less reliable in HIV co-infected persons, due to low humoral response <br> • VL-HIV co-infected patients have shown high sensitivity to detection of antigen in urine by the latex agglutination test. | [30–34, 40, 42] |

number of macrophages available for converting SSG to its active form leading to higher treatment failures among them.

Decrease in plasma glutathione (GSH), which is an important antioxidant, can affect both the immune response of the host and the mechanism of action of SSG with resultant increase in treatment failure with advancing age [67, 68]. When the proportion of treatment failures with age was tabulated with average plasma GSH levels that was adapted from Giustarini et al, 2006 (Fig 2, S1 Datasets) showed an interesting relationship.

In Fig 2, the linear forecast trend line of proportion of patients with treatment failure (red dotted line) increased in contrast to the decreasing linear forecast trendline of the average plasma glutathione (GSH) levels (black dotted line) with advancing age. The considerable drop seen in treatment failure and the corresponding drop in the GSH level in the age group 41–50 years as seen in Fig 2, could be related to menopause/andropause. The mean age of perimenopause is 46.1 (±3.7) years in Sri Lankan females while andropause begins by about 40 years, which would lead to oxidative stress enhancing parasite clearance and the drop in treatment failure seen in the same age group in this study [68].

Higher treatment failure seen in lesions on trunk (16/18, 89%) as opposed to those on head and neck (31/44, 70%) could also be explained by haemodynamic features such as the rich blood supply in scalp and face (as opposed to the trunk) which aids in healing. Furthermore, blood vessels with atherosclerotic plaques leading to less blood supply and interference of inflammatory cell migration through capillaries to the lesions and dermal atrophy could also account for higher treatment failure in elderly than in children.

The noticeably high treatment failure percentage in the study group (75.1%) is a matter of concern which calls for further research. In addition to the host and parasite related factors, extraneous factors such as chemicals might play a role in treatment response. Hambantota is a coastal district known for its agriculture and consumer preference for tuna. Even though the levels of arsenic in agrochemicals and fish seem to be much lower than the maximum contaminant levels and may not be significant or toxic when considered individually, it might be postulated that study participants may have been chronically exposed to sub-lethal levels of arsenic with the development of arsenic-resistant parasite strains that might be cross-resistant to antimony therapy [46, 68, 70–75]. Whether such environmental factors affect the treatment response in CL calls for further studies in different regions of the country. Excluding patients over 70 years of age, which accounts for approximately 4.7% of Sri Lankan population (www. statistics.gov.lk) may be viewed as a limitation.

According to estimates, nearly one third of the Sri Lankan population in 2018 live at risk of leishmaniasis, with the incidence rate of leishmaniasis in the Southern Province (the study location) showing an alarming increase from 1.2 cases / 100,000 in 2001 to 117.2 cases /

**Table 5. Mechanisms of the antimony resistance.**

| | Mechanism | Comment |
|---|---|---|
| 1 | Reduced uptake of the drug by the parasite [45–47]. | Aquaglyceroporin1 (AQP1) is known to facilitate the uptake of SbIII by the parasite |
| | | Downregulation of AQP1 was seen drug resistant parasites. |
| 2 | Increased intracellular thiol levels [47–49]. | In drug sensitive strains, SbIII disrupts the thiol homeostasis by: - a) inducing outflow of thiols and from parasites:<br>• Intracellular thiols such as trypanothione (TSH), glutathione (GSH) and cysteine, in *Leishmania* maintain the thiol redox homeostasis, protecting the parasite from chemical and oxidative stress.<br>• The γ-GCS gene encodes an enzyme catalyzing the rate limiting step of glutathione (GSH) biosynthesis and the ODC gene encodes for an enzyme regulating the biosynthesis of polyamines. Polyamines are the precursor metabolites of trypanothione.<br>• Antimony resistant strains have shown non-consistent upregulation of γ-GCS and over expression ODC genes, increasing the intra-cellular thiol-dependent antioxidant capacity, resulting in resistance to antimony.<br>and<br>b) inhibiting the reduction of trypanothione: Trypanothione reductase gene is amplified in antimony resistant isolates. This leads to high intracellular trypanothione levels and increased resistance to SbIII. |
| 3 | sequestration and rapid drug efflux [13, 45, 49, 50]. | ATP-binding cassette (ABC) transporters efflux drug out of the parasite or sequestrate the drugs in intracellular vesicles. |
| | | Eg: The two classes of ABC transporters, P-glycoprotein (eg: MRPA) and multi-drug resistance-related protein (eg: MRP1) known to lead to multi drug resistance. Genes encoding these transports have been amplified in antimony resistant parasites. |
| 4 | Altered membrane fluidity [51]. | Changes in membrane fluidity have been demonstrated in resistance to antimony combinations |
| 5 | Heat shock proteins and cell death related proteins [46, 52–55]. | Heat shock protein (eg: HSP83 and HSP70) associated modulation of cell death has been reported in resistant parasites. Cell death related protein tyrosine phosphate (PTP), proliferating cell nuclear antigen (PCNA) were upregulated and mitogen-activated protein kinase (MAPK) was downregulated in antimony resistant strains. |
| 6 | Modulation of host-pathogen interaction the host immune response [36, 56–58]. | Leishmania modulates signaling pathways of the host macrophages. |
| | | Drug resistant parasites have modulated the host pathogen interaction and there by the host immune response in previous studies. |
| 7 | Differentially expressed 8proteins associated with antimony resistance [59–62]. | Some proteins have been shown to be differentially expressed in with antimony response. Eg: Proteins such as the histone1, H2A, H4 and leucine-rich repeat protein are over expressed in antimony resistant parasites while proteins such as the kinetoplastid membrane protein (KMP-11) are under-expressed. |
| 8 | Misuse of the antimony drugs [30]. | Practices such as inadequate dose, inappropriate regimes, free availability, management of patients by unqualified persons and not completing treatment have led to development of subtherapeutic levels of antimony in blood causing development of parasite tolerance to antimony. |

100,000 in 2018 [6]. High rates of treatment failures as reported here may explain this rapid disease spread that threatens the plans for control and elimination of leishmaniasis in the region.

Of the two standard therapeutic measures (IL-SSG and cryotherapy) used in Sri Lanka, IL-SSG given until cure is the widely used first line treatment and the drug resistance is an

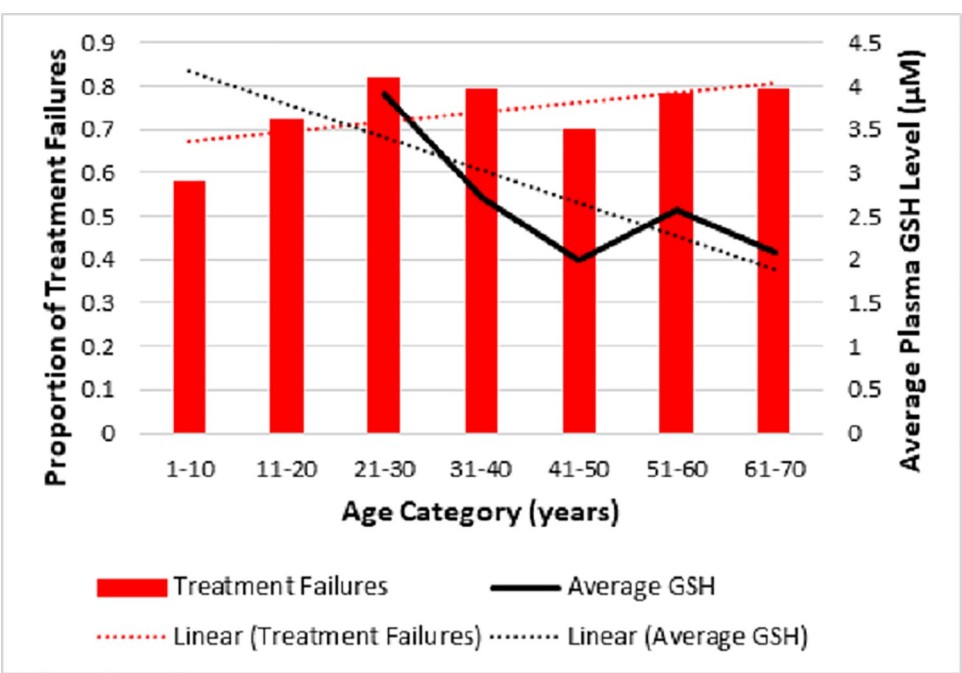

**Fig 2. Proportion of treatment failures versus age category with average plasma glutathione (GSH) levels adapted from Giustarini et al, 2006 [69].**

emerging problem. Even though cryotherapy is also used to a lesser extent, it has limitations such as inability to use on areas such as the face, scarring, keloid formation, ulceration and post inflammatory depigmentation specially in skin of colour [7, 76]. Intramuscular injection of SSG (IM-SSG) is not widely used for treatment of CL in Sri Lanka due to more side effects when compared with local infiltration. However, IM-SSG is used when the lesion is very large, many in number or when It's difficult to administer IL-SSG due to the location of the lesion. Even though cryotherapy is an accepted, relatively cheap standard therapy in Sri Lanka, it has limitations such as inability to apply on sites as such as face, disfigurement/scarring, keloid formation, depigmentation specially in skin of colour and smear positivity for long periods following treatment [76–78]. Therefore, it is extremely important to review the current treatment protocols and introduce safe and efficacious alternative treatment methods. A recent study has proven thermotherapy to be safe and efficacious for persons not healed by IL-SSG in Sri Lanka [17]. With further feasibility studies such treatment methods may be incorporated to the local guidelines as the first line treatment methods or alternative methods for treatment failures.

## Conclusions

In conclusion, this study characterises in detail the leishmaniasis treatment failures in Sri Lanka initiating and encouraging further studies focusing on emerging drug resistance. Nearly three fourths (75.1%) of the study population failed treatment with IL-SSG and majority of treatment failures were > 20 years of age (84%). Lesions located on trunk and nodules seemed least sensitive to antimony therapy. Factors associated with or having an impact on treatment failure varied among different age groups, which warrants further investigations. It would be timely to revisit and revise the current treatment strategies of CL in Sri Lanka. Understanding and combating CL treatment failure will aid the containment of the escalating CL spread in Sri

Lanka, with favourable impact on the regional and even global-level disease control and elimination efforts.

## Supporting information

**S1 Table. Binary logistic regression conducted with varying conditions and the respective results.**
(PDF)

**S1 Datasets. Datasets relevant to the manuscript.**
(XLSX)

## Acknowledgments

We are thankful to the Directors, Medical Officers, Nursing Officers and staff members of Dermatology Clinics of BH Tangalle and DGH Hambantota and the study participants. We thank Mr.Sudath Weerasingha and Ms.Yasasmi Gange for technical assistance. We are grateful to the Head and Staff of Department of Parasitology, Faculty of Medicine, University of Colombo.

## Author Contributions

**Conceptualization:** Hermali Silva, Nadira D. Karunaweera.

**Formal analysis:** Hermali Silva, Vasana Chandrasekara, Kalaivani Chellappan.

**Funding acquisition:** Hermali Silva, Nadira D. Karunaweera.

**Investigation:** Hermali Silva, Achala Liyanage, Theja Deerasinghe.

**Methodology:** Hermali Silva, Nadira D. Karunaweera.

**Resources:** Hermali Silva, Achala Liyanage, Theja Deerasinghe, Nadira D. Karunaweera.

**Supervision:** Nadira D. Karunaweera.

**Writing – original draft:** Hermali Silva.

**Writing – review & editing:** Vasana Chandrasekara, Kalaivani Chellappan, Nadira D. Karunaweera.

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
