## [Decision Letter · Decision Letter 0]

6 Sep 2021

PONE-D-21-23044Treatment failure to sodium stibogluconate in cutaneous leishmaniasis : A challenge to infection control and disease eliminationPLOS ONE

Dear Dr. Karunaweera,

Thank you for submitting your manuscript to PLOS ONE. After careful consideration, we feel that it has merit but does not fully meet PLOS ONE’s publication criteria as it currently stands. Therefore, we invite you to submit a revised version of the manuscript that addresses the points raised during the review process. Please revise the manuscript as per the suggestions from the reviewers and the Editor.  

We look forward to receiving your revised manuscript.

Kind regards,

Bhaskar Saha

Academic Editor

PLOS ONE

Journal Requirements:

2. We note that Figure 1 in your submission contain map/satellite images which may be copyrighted. All PLOS content is published under the Creative Commons Attribution License (CC BY 4.0), which means that the manuscript, images, and Supporting Information files will be freely available online, and any third party is permitted to access, download, copy, distribute, and use these materials in any way, even commercially, with proper attribution. For these reasons, we cannot publish previously copyrighted maps or satellite images created using proprietary data, such as Google software (Google Maps, Street View, and Earth). For more information, see our copyright guidelines: http://journals.plos.org/plosone/s/licenses-and-copyright.

Additional Editor Comments:

Antimony-based anti-leishmanial treatment has been in use for many decades. Such prolonged use leads to resistance of the target pathogen to the anti-leishmanial. This situation can become grave with lack of appropriate control policies.

While the manuscript deals with an important issue in Leishmania treatment, it requires a few modifications as raised by the reviewers. Please include the different mechanisms of the antimony resistance published till date in the form of a table. Another Reviewer also suggested to provide a table on prevalence, parasites and disease profiles including serological profiles. A comparison between CL and VL for antimony-resistance will be a good guideline for the readers.

Reviewers' comments:

Reviewer's Responses to Questions

**Comments to the Author**

1. Is the manuscript technically sound, and do the data support the conclusions?

Reviewer #1: Yes

Reviewer #2: Yes

2. Has the statistical analysis been performed appropriately and rigorously? 

Reviewer #1: Yes

Reviewer #2: Yes

3. Have the authors made all data underlying the findings in their manuscript fully available?

Reviewer #1: Yes

Reviewer #2: Yes

4. Is the manuscript presented in an intelligible fashion and written in standard English?

Reviewer #1: Yes

Reviewer #2: Yes

5. Review Comments to the Author

Reviewer #1: This is an interesting paper on Treatment failure to sodium stibogluconate in cutaneous leishmaniasis.

The study is well designed with sound analysis. However, the manuscript would be strengthened by addition of a table on clinical serological parameters.

Reviewer #2: The efforts taken by authors to report an crucial health issue in Shrilanka is appreciated. However study needs to be more thorough with respect to the molecular mechanisms associated with drug resistance which will help to discover the drugs that could be used to treat SSG resistant Leishmania strains.

6. PLOS authors have the option to publish the peer review history of their article (what does this mean?). If published, this will include your full peer review and any attached files.

Reviewer #1: **Yes: **Kalpana Pai

Reviewer #2: No

---

## [Author Response · Author response to Decision Letter 0]

3 Oct 2021

Response to Reviewers:

Itemized list of specific responses

PONE-D-21-23044

Treatment failure to sodium stibogluconate in cutaneous leishmaniasis : A challenge to infection control and disease elimination

PLOS ONE 

Please find below the itemized list of specific responses to each of the comments (page and line numbers refer to the unmarked version of the revised manuscript without tracked changes - the file labeled 'Manuscript').

[1] Journal requirements:

Response: PLOS ONE style template was referred, and the manuscript has been revised accordingly. 

2) We note that Figure 1 in your submission contain map/satellite images which may be copyrighted 

Response: Removed the Figure 1 from the manuscript since GPS coordinates have been included in the manuscript (page no: 5, line no: 116-117)

3) Please include captions for your Supporting Information files at the end of your manuscript, and update any in-text citations to match accordingly.

Response: Supporting information file renamed as ‘S1 Table’ , caption is at the end of the manuscript and in-text citations matched accordingly.

[2] Editor Comments:

Antimony-based anti-leishmanial treatment has been in use for many decades. Such prolonged use leads to resistance of the target pathogen to the anti-leishmanial. This situation can become grave with lack of appropriate control policies.

While the manuscript deals with an important issue in Leishmania treatment, it requires a few modifications as raised by the reviewers. Please include the different mechanisms of the antimony resistance published till date in the form of a table. Another Reviewer also suggested to provide a table on prevalence, parasites and disease profiles including serological profiles. A comparison between CL and VL for antimony-resistance will be a good guideline for the readers.

Response: 

1) Authors agree

2) Causative parasites, prevalence, clinical profile and serological diagnosis of the main disease forms are given as a table (Table 4). Comparison between CL & VL for antimony resistance is given in the manuscript text (page no: 16, line no: 243-252)

3) Different antimony resistance mechanisms are now summarized in table 5.

[3] Review comments to the Author:

Reviewer #1: This is an interesting paper on Treatment failure to sodium stibogluconate in cutaneous leishmaniasis.

The study is well designed with sound analysis. However, the manuscript would be strengthened by addition of a table on clinical serological parameters.

Response: Authors agree . Table 4 has been added with the relevant information. 

Reviewer #2: The efforts taken by authors to report an crucial health issue in Shrilanka is appreciated. However study needs to be more thorough with respect to the molecular mechanisms associated with drug resistance which will help to discover the drugs that could be used to treat SSG resistant Leishmania strains.

Response: Authors agree. The relevant information has been added in table 5 and described in the manuscript on page: 16, line: 252-256

---

## [Editor Report · Decision Letter 1]

11 Oct 2021

Treatment failure to sodium stibogluconate in cutaneous leishmaniasis : A challenge to infection control and disease elimination

PONE-D-21-23044R1

Dear Dr. Karunaweera,

We’re pleased to inform you that your manuscript has been judged scientifically suitable for publication and will be formally accepted for publication once it meets all outstanding technical requirements.

Kind regards,

Bhaskar Saha

Academic Editor

PLOS ONE
---

## [Editor Report · Acceptance letter]

13 Oct 2021

PONE-D-21-23044R1 

Treatment failure to sodium stibogluconate in cutaneous leishmaniasis : A challenge to infection control and disease elimination 

Dear Dr. Karunaweera:

I'm pleased to inform you that your manuscript has been deemed suitable for publication in PLOS ONE. Congratulations! Your manuscript is now with our production department. 

Kind regards, 

on behalf of

Dr. Bhaskar Saha 

Academic Editor

PLOS ONE